# Soil Quality Assessment and Management in Karst Rocky Desertification Ecosystem of Southwest China

Qian Wu [1], Wei Zheng [2,3,*], Chengjiao Rao [2,3], Enwen Wang [1] and Wende Yan [2,3]

[1] Faculty of Resources and Environmental Engineering, Anshun University, Anshun 561000, China
[2] Faculty of Life Science and Technology, Central South University of Forestry & Technology, Changsha 410004, China
[3] National Engineering Laboratory for Applied Technology of Forestry & Ecology in South China, Changsha 410004, China
* Correspondence: emailofzhengwei@163.com; Tel.: +86-0731-85623482

**Abstract:** Karst rocky desertification is a common phenomenon in terrestrial ecosystems, and the deterioration of soil quality has a serious side effect on the aboveground vegetation and underground environmental factors. To clarify the variety of soil quality in different rocky desertification grades in typical karst areas of southwest China, the soil quality of four rocky desertification grades was calculated by a single model (SQI: soil quality index), two screening processes (TDS: total dataset and MDS: minimum dataset) and three scoring methods ($SSF$: standard scoring function, $S_L$: linear scoring function and $S_{NL}$: nonlinear scoring function). The key results are as follows: Significant differences were found in the soil environment factors in non-rocky desertification (NRD), light rocky desertification (LRD) and moderate rocky desertification (MRD) as compared to intense rocky desertification (IRD) ($p < 0.01$). Except for total potassium (TK), manganese (Mn) and amylase, the other soil environmental factors showed U-shaped changes. In contrast, TK, Mn and amylase increased first and then decreased. Additionally, the SQI based on MDS in $SSF$, $S_L$ and $S_{NL}$ was IRD (0.58) > NRD (0.48) > LRD (0.45) > MRD (0.43), IRD (0.53) > NRD (0.42) > LRD (0.39) > MRD (0.36) and IRD (0.57) > NRD (0.47) > MRD (0.42) > LRD (0.40), respectively. However, the SQI was always in the trend of IRD > NRD > MRD > LRD based on the TDS. Overall, although the soil area is scarce, the edaphic properties, enzyme activities and soil quality are not poor in the IRD. Furthermore, we found that $S_{NL}$ was more suitable for the evaluation of soil quality in the karst rocky desertification area ($R^2 = 0.63$, $p < 0.001$ and the coefficient of variation = 30.69%). This research helps to clarify the variation in soil properties and quality during the succession of rocky desertification and provides guidelines for the sustainable management of soil quality in areas of southwest China.

**Keywords:** karst rocky desertification; soil quality and properties; karst shallow fissure; intense rocky desertification

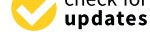



## 1. Introduction

Karst rocky desertification (KRD), which is caused by soil erosion and irrational, intensive land use in fragile karst geoecological environments has become a severe environmental and social issue throughout the world [1]. Generally, areas with severe rocky desertification have high bedrock exposure rates, serious shortages of effective cultivated areas and thin soil layers [2]. In contrast, areas with light rocky desertification have high vegetation coverage and a wider effective cultivated area of soil [3]. Thus, ecological degradation is usually characterized by the loss of cultivated soil, water shortages, soil erosion, decreased biodiversity and phytocommunity degradation in KRD areas [4,5]; this results in uneven land resources and soil quality under different rocky desertification grades. This is the main reason for the aggravation of the problem in the KRD area of southwest China over many years [6].

The global karst landform covers approximately 22 million square kilometers and represents approximately 12% of the Earth's surface [7,8]. In China, karst ecosystems cover more than 900 thousand square kilometers and represent nearly 1/10 of the total land area [9]. Southwest China is one of the three major continuous KRD areas in the world, and the KRD areas are mainly distributed in eight provincial areas (Guizhou, Yunnan, Sichuan, Guangxi, Hunan, Hubei, Chongqing and Guangzhou) [10,11]. Guizhou Province has the largest distribution area and is the most seriously affected by rocky desertification, which creates substantial obstacles and threats to economic development and the ecological environment [12]. Due to the interaction of natural environmental factors (such as precipitation, temperature and the differences in land use types) with man-made activities, the habitats suffer from further soil erosion, large-area bedrock exposure and degradation of land productivity [13]. Therefore, the composition of the plant community is becoming increasingly singular, the vegetation coverage is becoming lower and the exposure rate of bedrock is increasing, which further leads to more serious water and soil loss and leakage [14,15].

The complete definition of soil quality has been described by Karlen et al. [16] and the committee for the Soil Science Society of America as 'the capacity of soil to function to sustain plant and animal productivity, to maintain or enhance water and air quality and to support human health and habitation'. Many evaluation methods have been used to assess soil quality. The most common methods to calculate soil quality are soil quality index methods (SQI) [17,18] and geostatistical methods [19,20] because they are easy to use and quantitatively flexible [20]. Additionally, they can be used as valid support in the assessment of the soil ecosystem and its management by combining temporal and spatial information [21]. Thus, when we use these methods to calculate the SQI of the KRD area, the selection of soil environmental factors is particularly important. Soil quality indicators are the comprehensive performance of physical, chemical and biological properties, which can clearly respond to the variety of soil conditions in terrestrial ecosystems [18]. At present, in the process of SQI calculation, the total dataset (TDS) and minimum dataset (MDS) have been widely used for the selection of soil quality factors [17]. However, few people use different methods to calculate soil quality in different KRD areas. Hence, it is necessary to evaluate soil quality under different rocky desertification grades by reasonable methods in southwest China, and we hypothesized that the increase in rocky desertification has a side effect on soil quality in the karst ecosystem and is expected to provide a scientific basis for soil quality management in the KRD area.

Due to the different evolution processes of rocky desertification with large variations in the soil area, soil erosion, vegetation coverage and soil properties of different rocky desertification grades, soil management is difficult in the KRD area of southwest China [22]. For these reasons, the goals of this research were (i) to evaluate soil quality under different rocky desertification grades in southwest China by using a single soil quality model (SQI), two screening processes (TDS and MDS) and three scoring methods ($SSF$, $S_L$ and $S_{NL}$); (ii) to determine which method is most suitable for soil quality evaluation in the KRD area; and (iii) to quantify the relationship between rocky desertification grade and soil quality.

## 2. Materials and Methods

### 2.1. Study Region

We carried out this study in Puding County, Anshun City, Guizhou Province, southwest China (26°17′12″~26°22′09″ N, 105°44′53″~105°45′19″ E) (Figure 1). The study site is located in the watershed between the Yangtze River system and the Pearl River system on the Guizhou Plateau. This is a typical karst area in the warm and humid climate area of the mid-subtropical monsoon, with an annual average temperature of 15 °C and an annual temperature range of about −3.4–34.8 °C. The mean annual rainfall is 1300 mm, and 83%–88% of the total annual rainfall (891–1390 mm) is concentrated from May to October [23]. In addition, the average altitude is 1387 m, the altitude range is 1100–1600 m and the mean slope is 24° [23,24]. The main crops and other economic crops include rice,

corn, sorghum, wheat, loquat, plum, flue-cured tobacco and peanuts. Additionally, the soil types with the broadest distributions are yellow, lime and paddy soil, accounting for 37.18%, 33.24% and 27.02% of the total cultivated land area, respectively [25].

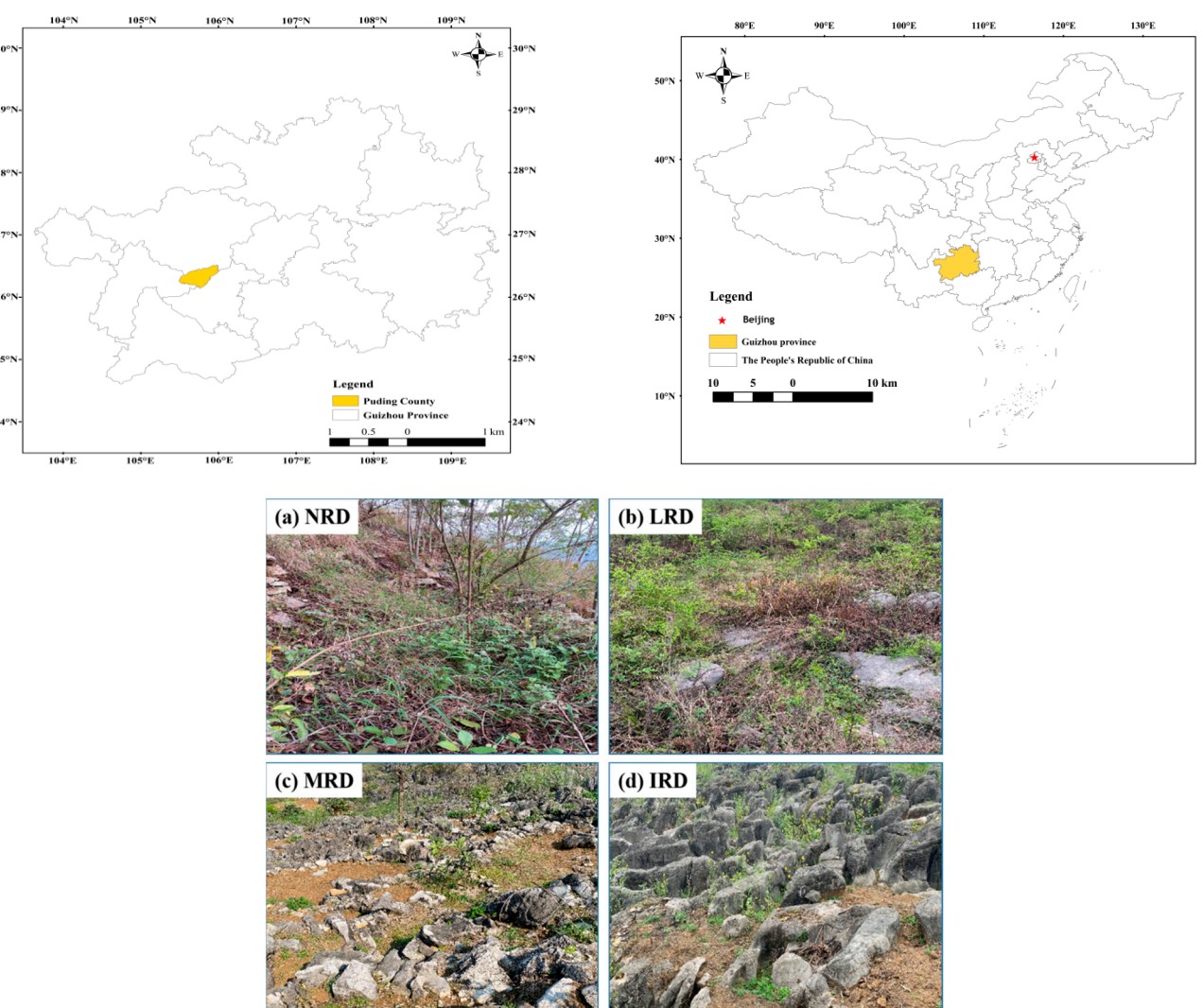

**Figure 1.** Location of the study area and sampling sites in Puding county, Guizhou province and the pictures of different rocky desertification grades in the study: (**a**) non rocky desertification (NRD), (**b**) light rocky desertification (LRD), (**c**) moderate rocky desertification (MRD) and (**d**) intense rocky desertification (IRD).

### 2.2. Soil Sampling Measurements

In March 2021, soil sample plots of four different rocky desertification grades (NRD: non rocky desertification, LRD: light rocky desertification, MRD: moderate rocky desertification and IRD: intense rocky desertification) were selected based on the industry standard of the State Forestry Administration (LY/T 1840-2009) and the classification standard of rocky desertification (Table 1) [26,27]. Additionally, we recorded the basic information of each rocky desertification sample (Table 2). Then, three small quadrats of 50 × 50 m were selected for the four different rocky desertification grades. The distance between each small quadrat was at least 200 m. Finally, soil samples were collected from the 0–10, 10–20, and 20–30 cm depths at each plot and every sample was a mixture of five subsamples collected in an 'S' shape. The total number of soil samples collected was 4 treatments × 3 soil depths × 3 replicate sites × 5 subsamples = 180. In the laboratory, except for the soil enzyme activity which was sieved by 2 mm and placed in a 4 °C refrigerator, the other

samples were air-dried and sieved to 0.149 mm for physicochemical analyses, respectively. Soil physicochemical properties are indicators that reflect the basic properties of soil but are also the key indicators that reflect soil fertility. Thus, in this study, we referred to the indicators selected by previous researchers to calculate soil quality combined with a correlation analysis of soil environmental factors under different rocky desertification grades and selected a total of ten indicators as the TDS. These soil parameters included sucrase, amylase, β-glucosidase, total nitrogen (TN) [28], total phosphorus (TP) [29], total potassium (TK) [30], pH, soil organic matter (SOM) [31], heavy metal manganese (Mn) [30] and soil moisture content (SMC).

**Table 1.** Classification standard of different rocky desertification grades in the study: non rocky desertification (NRD), light rocky desertification (LRD), moderate rocky desertification (MRD) and intense rocky desertification (IRD).

| Rocky Desertification Grade | 0.2 km² Bare Rock Rate (%) | 0.2 km² Vegetation + Soil Coverage | Degree of Soil Erosion | Vegetation Characteristics | Average Soil Depth |
|---|---|---|---|---|---|
| NRD | 20–30 | 70–80 | not obvious | dominated by rocky and xerophytic shrubs | shallow |
| LRD | 31–50 | 50–69 | relatively obvious | sparse shrub and grass | shallow |
| MRD | 51–70 | 30–49 | obvious | low structure, coverage and biomass are relatively stable | shallow |
| IRD | 71–90 | 10–29 | strong | mainly shrub grass of low structure | shallow |

**Table 2.** Basic information of four rocky desertification grade samples in the study: non rocky desertification (NRD), light rocky desertification (LRD), moderate rocky desertification (MRD) and intense rocky desertification (IRD). Slope direction (SD), rocky exposure rate (RER) and vegetation coverage (VC).

| Plot Type | SD | Slope/° | Altitude/m | RER/% | VC/% | Interference Conditions |
|---|---|---|---|---|---|---|
| NRD | Northeast | 23 | 1300–1305 | 26 | 92 | Wasteland, without interference |
| LRD | Northeast | 20 | 1202–1225 | 42 | 79 | Wasteland, without interference |
| MRD | Northeast | 20 | 1162–1289 | 66 | 58 | Light human disturbance |
| IRD | Northeast | 16 | 1158–1160 | 85 | 20 | Intense human disturbance |

*2.3. Soil Quality Evaluation*

All selected soil environmental indicators, which are called the TDS, had a significant impact on the results of the soil quality evaluation [32]. The following steps were used to calculate soil quality. First, representative indicators were identified. Second, through the dimension reduction analysis method in principal component analysis (PCA), multiple indices were transformed into several irrelevant comprehensive indices, and eigenvalues ≥ 1 were retained as the principal components. Third, we rotated the maximum communalities and selected the high load indicators in the rotated load matrix to confirm the MDS. Finally, MDS was determined. The determination of the MDS mainly includes the following steps: (1) the TDS are divided into groups, taking the characteristic value ≥ 1 as the extraction principle, several principal components are obtained through PCA; (2) the indicators with load values > 0.5 are divided into a group, and if there are indicators with

load values > 0.5 in different principal components, they are divided into groups with low correlations according to the correlation analysis between the indicators; (3) the indicators with norm values > 90% in each group are selected. The calculation of the norm value (5) [33] can avoid the redundancy of soil data as much as possible and retain its soil index information to the greatest extent. Provided that there was no correlation, they all can be selected for the MDS. In contrast, the largest norm value is selected to enter the MDS.

After the MDS was determined, the SQI was evaluated (4) [32,34] according to the standard scoring function (*SSF*) (1) [32,35], linear regression equation (*S$_L$*) (3) and nonlinear regression (*S$_{NL}$*) (2) [33]. Finally, the SQI was calculated as follows:

$$SSF = \begin{cases} 1 & x < L \\ 1 - 0.9\dfrac{x-L}{U-L} & L \le x \le U \\ 0.1 & x > U \end{cases} \tag{1}$$

where $x$ is the actual measured indicator value, and $L$ and $U$ are the lower and upper threshold values, respectively [32,35].

$$S_{NL} = \frac{a}{1 + \left(\frac{x}{x_0}\right)^b} \tag{2}$$

$$S_L = \frac{x-L}{U-L}\ldots(i) \quad S_L = 1 - \frac{x-L}{U-L}\ldots(ii) \tag{3}$$

where $x_0$ is the average value of the indicator value, $a$ is the maximum score ($a = 1$) reached by the function and $b$ is the equation's slope value. When $b$ is $-2.5$ and $2.5$, the curve was set for 'more is better' (i) and 'less is better' (ii), respectively [34].

$$SQI = \sum_{i=1}^{n}(S_i W_i) \tag{4}$$

where $S_i$ is the score of each indicator and is obtained by the product of weight and communality, $W_i$ is the ratio of the communality of the principal component molecule to the total communalities and n is the total number of the MDS [35].

$$N_{jk} = \sqrt{\sum_{1}^{k}\left(u_{jk}^2 \lambda_k\right)} \tag{5}$$

where $N_{jk}$ is the norm value, $k$ is the number of eigenvalues $\ge 1$ in the principal component, $\lambda_k$ is the eigenvalue of the $k^{th}$ principal component and $u_{jk}$ is the single factor load of the $j$ index [33].

### 2.4. Data Analysis

All analyses were conducted in SPSS 22 (SPSS Inc., Chicago, IL, USA). Excel 2010 was used to analyze all data and fitted linear regression equations, respectively. All figures were analyzed in Origin 2021 pro and GraphPad Prism 8.

## 3. Results

### 3.1. Eco-Environmental Factors

We comprehensively selected ten soil environmental factors as the TDS, including SMC, TN, TP, TK, SOM, Mn, pH, sucrase, amylase and β-glucosidase. As shown in Figure 2, the edaphic properties (TN, SOM, TP) from the NRD to the IRD showed 'U-shaped' changes. Under the same KRD grade, they all decrease with increasing soil depth. There was a significant difference ($p < 0.01$) between the different rocky desertification grades, and there was no difference between the different soil layers of the same grade ($p > 0.05$). However, the concentrations of Mn were the highest in MRD ($1.33 \pm 0.12$ g kg$^{-1}$), the pH values were

the highest in NRD and TK first increased and then decreased. There was no significant difference in SMC among the three grades of NRD, LRD and MRD, but the content of IRD (0–10 cm:25.85 ± 1.47, 10–20 cm: 26.02 ± 4.12 and 20–30 cm: 27.59 ± 3.69%) was much higher than that in the first three rocky desertification areas. As shown in Figure 3, the three soil enzyme activities decreased with the increase in the soil layer, sucrase increased with the increase in rocky desertification degrees, amylase from NRD to IRD showed LRD > MRD > IRD > NRD, while β-glucosidase first decreased and then increased, with significant differences among different rocky desertification grades ($p < 0.01$).

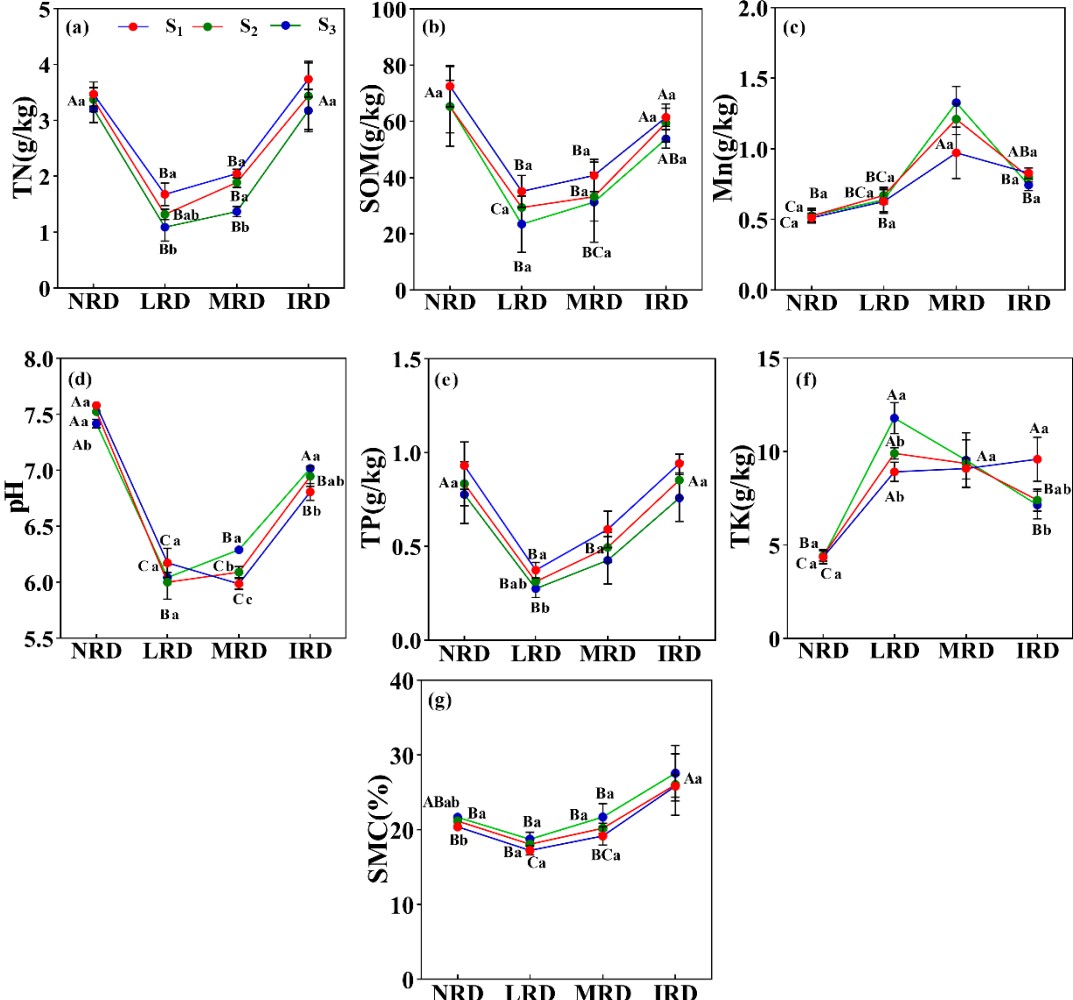

**Figure 2.** Concentrations of environmental factors in different rocky desertification grades in the study: non rocky desertification (NRD), light rocky desertification (LRD) and moderate rocky desertification (MRD) to intense rocky desertification (IRD). Different capital letters represent different rocky desertification grades of the same soil layer have significant differences, and different small letters represent significant differences between different soil layers of the same rocky desertification grade ($p < 0.05$). $S_1$: 0–10 cm, $S_2$: 10–20 cm and $S_3$: 20–30 cm. (**a**) Total nitrogen, (**b**) Soil organic matter, (**c**) Manganese, (**d**) pH, (**e**) Total phosphorus, (**f**) Total potassium and (**g**) Soil moisture content.

Soil enzyme activity is greatly affected by pH, SMC, Mn and TK (Figure 4). Sucrase was negatively correlated with pH and TK, and the correlation coefficients were −0.24 and −0.043, respectively. β-glucosidase was greatly affected by pH and the correlation coefficient was −0.025. From all of the soil environmental factors, pH had a strong negative correlation with SMC, TN, TP and SOM, while TN, TP, SMC and SOM were positively correlated. In addition, TK and Mn had a strong response to soil environmental factors in

TDS. Overall, soil enzymes are obviously affected by soil physical and chemical factors in rocky desertification areas.

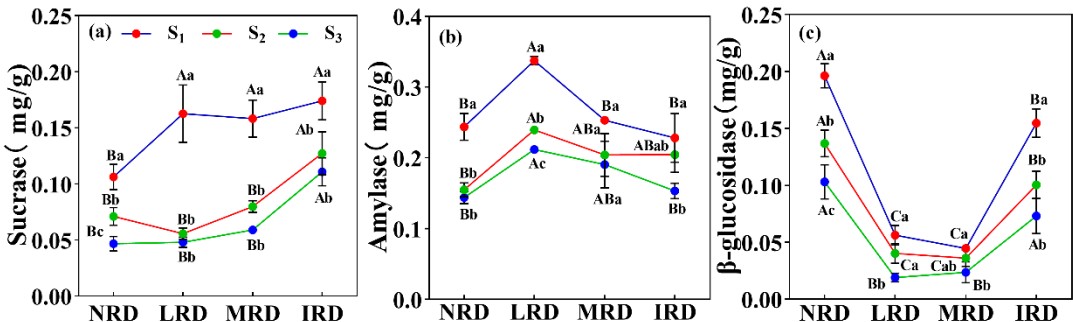

**Figure 3.** Soil enzyme activities in different (karst rocky desertification) KRD grades. Different capital letters represent different rocky desertification grades of the same soil layer have significant differences, and different small letters represent significant differences between different soil layers of the same rocky desertification grade ($p < 0.05$). (**a**) Sucrase, (**b**) Amylase and (**c**) β-glucosidase.

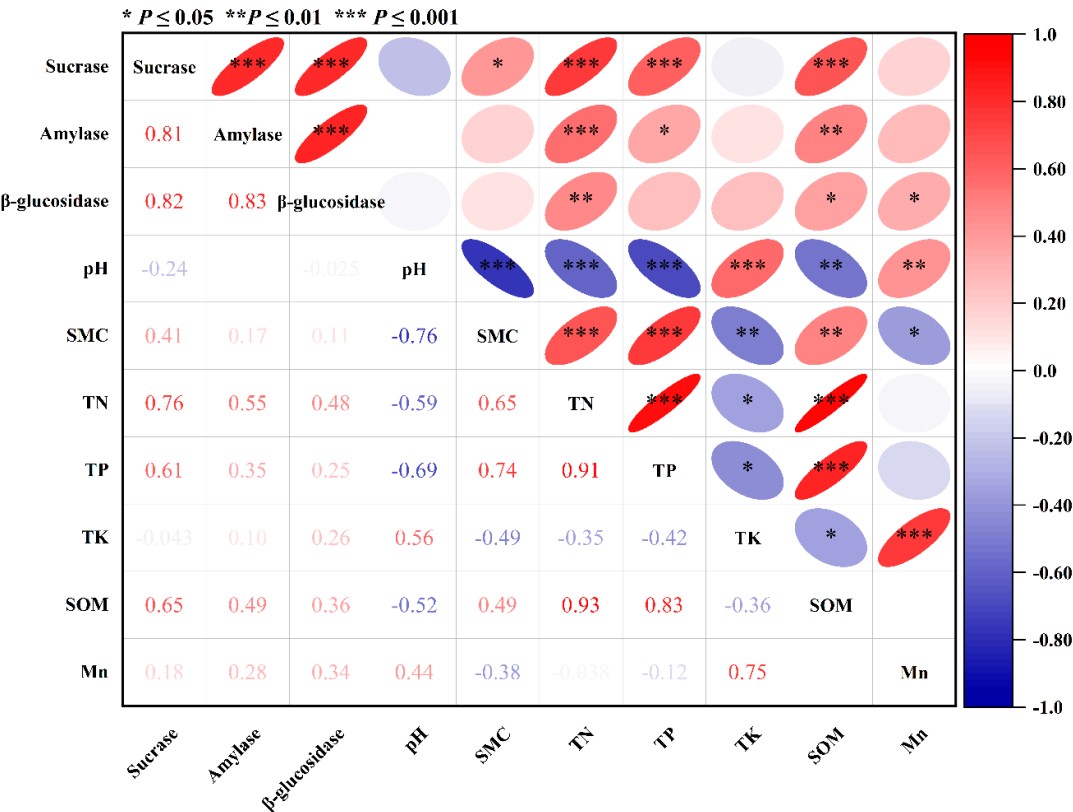

**Figure 4.** Correlation analysis of environmental factors and soil enzyme activities.

### 3.2. Soil Quality Based on Total Dataset Method

The dimension reduction analysis of PCA was used to analyze the TDS indicators (Table 3). Three main components were obtained, namely SOM, sucrase and amylase, which represented 94.14% of the ten eco-environmental factors. Among the three principal components, the loading factors were 0.987, 0.941 and 0.715. In the first principal component (PC1), the load factors of pH (0.940), TN (0.961) and TP (0.949) were within 10% of the maximum load factor, but it can be clearly seen in Figure 4 that these indicators had a strong correlation with SOM ($p < 0.01$). Thus, only SOM was selected in PC1 for the MDS. Similarly, the second (PC2) and third principal components (PC3) were sucrase and amylase in the MDS, respectively.

**Table 3.** Principal component analysis (PCA) load matrix of soil environmental factors.

| TDS | PC 1 | PC 2 | PC 3 |
|---|---|---|---|
| Sucrase | 0.125 | **<u>0.941</u>** | −0.248 |
| Amylase | −0.463 | 0.502 | **<u>−0.715</u>** |
| β-glucosidase | 0.897 | 0.149 | −0.315 |
| pH | 0.940 | −0.315 | −0.033 |
| SMC | 0.600 | 0.330 | 0.650 |
| TN | 0.961 | 0.246 | 0.065 |
| TP | 0.949 | 0.276 | 0.084 |
| TK | −0.851 | 0.277 | 0.193 |
| SOM | **<u>0.987</u>** | 0.084 | −0.092 |
| Mn | −0.463 | 0.575 | 0.572 |

The communalities of TDS and MDS were obtained from the PCA, and the weight of each indicator was calculated as shown in Table 4. Then, the three scoring functions ($SSF$, $S_L$ and $S_{NL}$) were used to obtain the scores of TDS under the four rocky desertification grades (Figure 5a). The SQI scores calculated by the three scoring functions showed the same trend: IRD > NRD > MRD > LRD, and using the $SSF$, $S_L$ and $S_{NL}$ scores from NRD to IRD were 0.56, 0.36, 0.41, 0.63, 0.51, 0.29, 0.35, 0.59, 0.55, 0.37, 0.39 and 0.58.

**Table 4.** Communalities, weight and norm value of environmental factors based on minimum dataset (MDS) and total dataset (TDS), respectively. COM: communalities.

| Indicator | TDS | | MDS | | |
|---|---|---|---|---|---|
| | COM | Weight | COM | Weight | Norm Value |
| Sucrase | 0.963 | 0.102 | 0.923 | 0.331 | 1.37 |
| Amylase | 0.977 | 0.104 | 0.911 | 0.327 | 1.59 |
| β-glucosidase | 0.926 | 0.098 | | | 2.24 |
| pH | 0.983 | 0.104 | | | 2.36 |
| SMC | 0.892 | 0.095 | | | 1.73 |
| TN | 0.988 | 0.105 | | | 2.38 |
| TP | 0.985 | 0.105 | | | 2.36 |
| TK | 0.839 | 0.089 | | | 2.14 |
| SOM | 0.989 | 0.105 | 0.955 | 0.342 | 2.43 |
| Mn | 0.871 | 0.093 | | | 1.55 |

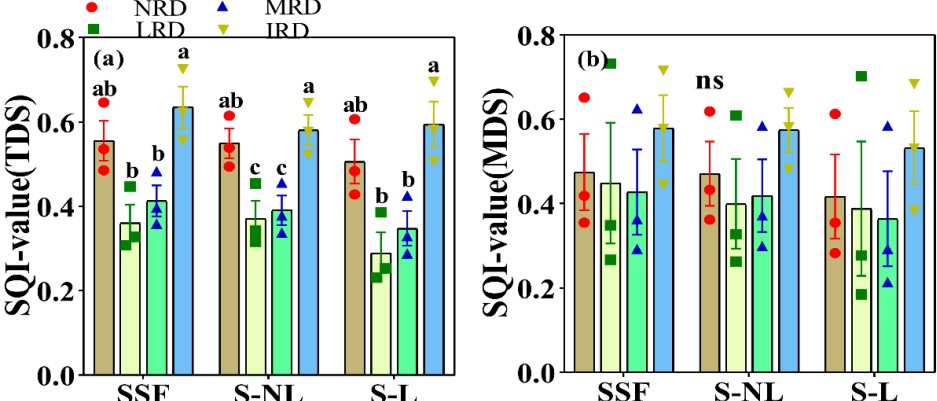

**Figure 5.** Score of soil quality index based on MDS and TDS in the study: soil quality index (SQI), total data set (TDS), minimum data set (MDS), standard scoring function ($SSF$), linear scoring function ($S_L$) and nonlinear scoring function ($S_{NL}$). Different small letters represent significant differences between different soil layers of the same rocky desertification grade ($p < 0.05$), ns: not significant. (**a**) Soil quality index according to the total data set and (**b**) Soil quality index according to the minimum data set.

### 3.3. Soil Quality Based on Minimum Dataset Method

We obtained communalities of 0.955, 0.923 and 0.911 for SOM, sucrase and amylase respectively, from the dimension reduction analysis of PCA (Table 4). Additionally, the weight values for SOM, sucrase and amylase were 0.331, 0.327 and 0.342, respectively. In the MDS, the three communalities indicated that the interpretation of SOM for the original environmental factors was the highest and the overall representation was the best, followed by sucrase and amylase.

As shown in Figure 5b, the scoring trend of soil quality obtained by the three scoring functions was different from that of TDS. The trend under SF and $S_L$ was IRD > NRD > LRD > MRD, and in $S_{NL}$ it was IRD > NRD > MRD > LRD. Additionally, the soil quality scores in $SSF$ for NRD, LRD and IRD were 0.48, 0.45, 0.43 and 0.58, respectively; the $S_L$ values were 0.42, 0.39, 0.36 and 0.53, respectively; and the $S_{NL}$ values were 0.47, 0.40, 0.42 and 0.57, respectively. This result indicated that the more serious the rocky desertification grade is, the better the soil quality is. In contrast, the lighter the rocky desertification grade is, the worse the soil quality is.

### 3.4. Soil Quality Index Validation

According to the criteria for dividing the score range of the SQI [36,37], as shown in Table 5 based on the $S_{NL}$, the excellent proportion of soil quality was 0, the good proportion was 27.78%, the medium proportion was 30.56% and the poor proportion was 41.67%. Furthermore, the excellent, good, medium and poor proportions of soil quality were 2.78%, 22.22%, 22.22% and 52.78%, respectively, in $S_L$. Similarly, the excellent, good, medium and poor values were 5.56%, 30.56%, 25% and 38.89%, respectively, in $SSF$. Additionally, the proportion of the three scoring functions was different, but the overall trend is the same based on the MDS, which showed that the ecological landscape area of poor soil quality in the KRD area was widespread. As shown in Table 5, the ranges of change of the SQI under the $S_{NL}$, $S_L$ and $SSF$ score functions were 0.214–0.709, 0.124–0.782 and 0.211–0.823, respectively. The mean values of the SQI from $S_{NL}$ to $SSF$ were 0.466, 0.425 and 0.483, respectively. In addition, the coefficient of variation from $S_{NL}$ to $SSF$ showed a trend $S_L$ (44.94%) > $SSF$ (35.61%) > $S_{NL}$ (30.69%). This result indicated that $S_{NL}$ is more suitable for soil quality evaluation in the KRD area.

**Table 5.** Soil quality statistics in the minimum dataset (MDS).

| Soil Quality Index | Change Range | Mean | Standard Deviation | Coefficient of Variation/% | Proportion of Sample Plots/% | | | |
|---|---|---|---|---|---|---|---|---|
| | | | | | Excellent (0.8–1.0) | Good (0.6–0.8) | Medium (0.4–0.6) | Poor (0–0.4) |
| SQI-NL | 0.214–0.709 | 0.466 | 0.143 | 30.69 | 0 | 27.78 | 30.56 | 41.67 |
| SQI-L | 0.124–0.782 | 0.425 | 0.191 | 44.94 | 2.78 | 22.22 | 22.22 | 52.78 |
| SQI-*SSF* | 0.211–0.823 | 0.483 | 0.172 | 35.61 | 5.56 | 30.56 | 25.00 | 38.89 |

Finally, the linear regression equation was used to fit the soil quality scores of the MDS and the TDS, and we found that the linear regression equation based on the $SSF$ ($R^2 = 0.52$) was not different from the $S_L$ (Figure 6). However, the $R^2$ of the linear regression equation was 0.63 in the $S_{NL}$. This further indicated that the $S_{NL}$ is more suitable for the KRD area than the $SSF$ and $S_L$ when we calculated the SQI based on three scoring functions under the MDS. Meanwhile, the proportions of excellent, good, medium and poor soil quality for MDS were 2.8%, 26.9%, 25.9%, and 44.4%, respectively, and for TDS were 0%, 19.4%, 45.4% and 35.2%, respectively (Figure 7).

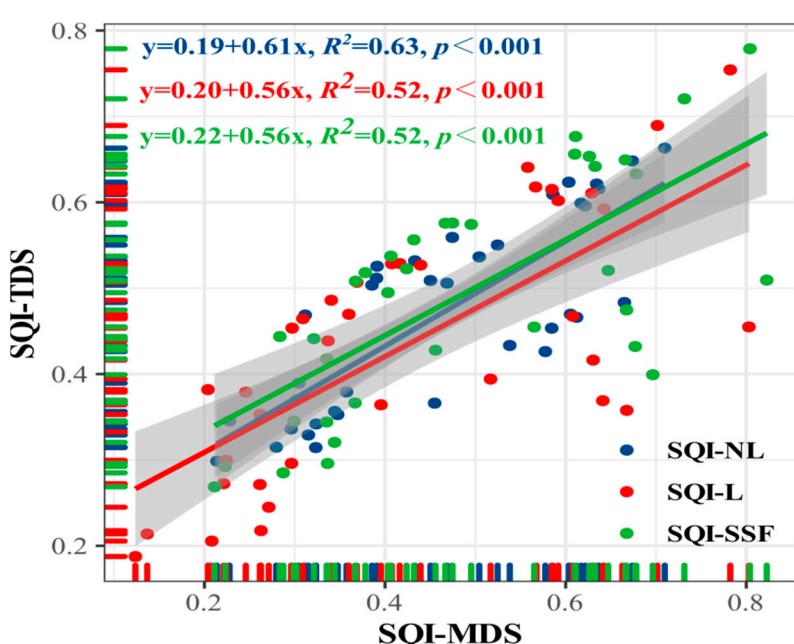

**Figure 6.** Analysis of soil quality index of the minimum dataset (MDS) and total dataset (TDS) by three scoring methods.

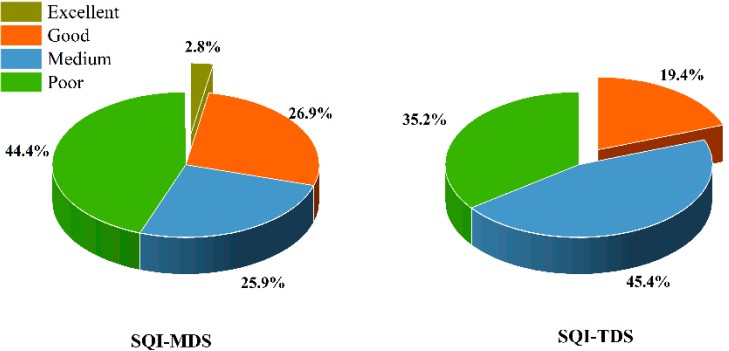

**Figure 7.** The proportion of soil quality in the minimum dataset (MDS) and total dataset (TDS).

## 4. Discussion

### 4.1. Soil Quality and Eco-Environmental Factors

Soil provides the nutrients and habitat necessary for plant diversity and soil microbial diversity in the KRD area [38]; a favorable soil environment is essential to supporting root growth and may have a positive impact on physicochemical properties [39,40]. Soil physical, chemical and biological indicators can reflect the state of the soil microenvironment and provide a theoretical basis for the restoration of rocky desertified ecosystems [41]. Many studies have measured the concentration and distribution characteristics of the soil pH value, TN, TP, TK, SOM, some heavy metals and many soil enzymes in karst areas [34,42]. Some reports showed that except for soil pH values and Ca, which increased with rocky desertification grade, the soil component contents were MRD > LRD > IRD [3,43]. However, the trend was IRD > NRD > LRD > MRD in our research. The main causes for this may be the differences in climate types, soil texture types and terrain conditions. We found that the areas with severe rocky desertification were the most seriously disturbed by human beings [34,44], which led to better performance of soil nutrients (pH, TN, TP, TK, SOM) and soil quality in IRD habitats [45,46]. Thus, there is an interesting phenomenon in the IRD area. Rocky desertification is very serious, but the soil properties are good. Of course, different karst types, topographic conditions, regional rainfall and temperature in rocky desertification areas are also important factors affecting soil nutrients [47].

In the soil ecosystem, enzyme activity is closely related to microbial community characteristics (such as soil bacteria and fungi) and linked to soil organic carbon (β-glucosidase and sucrase), TN (urease and nitrate reductase), and TP (alkaline/acid phosphatase) cycling [12,15]. In KRD areas, the decline in soil enzyme activities is significantly related to the rocky desertification grade [12,48]. In our study, the variation in soil enzyme activity was obvious with different rocky desertification grades. Partial experiments have revealed that the microbial characteristics and soil enzyme activity decline with an increase in the proportion of bedrock exposure rate under different rocky desertification conditions [24,38]. This is inconsistent with our research results, which may be because our IRD area has been seriously disturbed by human beings, which affects the microbial activity and plant diversity distribution and makes the soil enzyme activity higher in IRD. In addition, Bates et al. (2010) and Steven et al. (2015) [49,50] reported that KRD does not erode *ectomycorrhizal* fungal species richness but rather alters the microbial community. Although KRD reduced soil fertility and altered microbial community structures, microbial diversity did not diminish [51,52]. These studies further confirmed that the change in microbial activity in severe rocky desertification areas is an important factor affecting soil enzyme activity.

However, in our study soil enzyme activities and many environmental factors were highest in the IRD area, which is consistent with previous research results for karst shallow fissures in KRD ecosystems [38]; the soil properties in karst shallow fissures with severe rocky desertification are higher than those in areas with light rocky desertification [12,15]. We found that the content of soil nutrients in the IRD area is close to that in karst shallow fissures, and the large area of bedrock exposure in the IRD area is the basis for the formation of karst shallow fissures. Coupled with natural nitrogen deposition, rainfall flow convergence, etc. [53,54]. Therefore, we hypothesize whether the severe rocky desertification area of karst ecosystems in southwest China will develop into a large area of karst shallow fissures in the future (Figure 8). However, this needs to be confirmed by further research and observation. Overall, there are some differences in the study of different rocky desertification soil nutrients and enzyme activities in different regions that may be mainly due to the comprehensive influence of various factors, such as terrain slope, land use change, climate type, plant diversity, bedrock exposure rate, rain erosion and soil erosion [5,43].

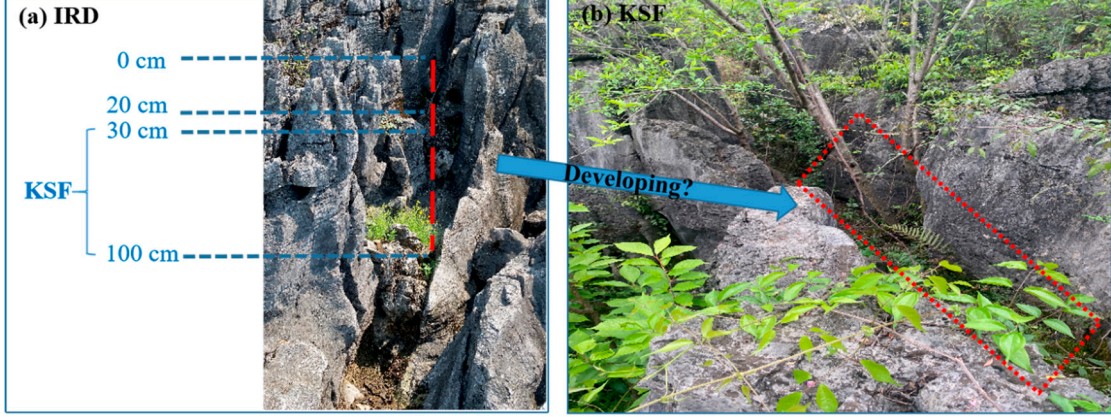

**Figure 8.** Comparison between intense rocky desertification and karst shallow fissure, (**a**): intense rocky desertification (IRD), (**b**): karst shallow fissure (KSF).

*4.2. Soil Quality Evaluation Methods*

Soil quality evaluation is of great importance to the comprehensive evaluation of soil ecosystems [34] and the method is broadly used because of its convenient operation, and reliable and accurate results [55]. In the calculation of SQI, TDS is an important method to select soil indicators, because it can obtain more comprehensive results, but too many soil indicators will make the data redundant [32]. Compared to TDS, using the MDS method to assess the SQI can significantly shorten the time and reduce data redundancy [55,56]. Thus, this study uses MDS and TDS to evaluate the SQIs of different rocky desertification grades.

The *SSF* is a common method for calculating the SQI, which can score in combination with the selected soil indicators to more intuitively reflect soil quality status [32]. However, in order to compare and screen more suitable methods for soil quality scoring in rocky desertification areas, we cite $S_L$ and $S_{NL}$, which can dimensionless soil index and convert it into a score of 0–1 for evaluation [33,57]. Through the dimension reduction analysis in PCA, screening the MDS can effectively reduce data redundancy. SOM can increase the content of soil nutrients and affect the structure and activity of the microbial community, which is a significant effect on soil fertility and plant community composition [58]. In addition, soil enzyme activity is the main indicator of indirect reactions of microbial activity characteristics and soil nutrient cycles, such as amylase, sucrase and β-glucosidase which are closely related to the cycle of soil organic carbon [17,59]. In our study, amylase and sucrase had higher percentage contributions, which indicated that soil organic carbon, SOM and microorganisms in rocky desertification areas are important soil quality indicators. SMC also has direct effects on soil properties and microbial communities and seriously affects the degradation of organic matter [60,61]. As an important factor limiting plant growth and development in the ecosystem, the effectiveness of TN, TP and TK is also particularly important to the ecosystem function and soil quality [62].

Scoring methods $S_L$ ($R^2 = 0.52$), $S_{NL}$ ($R^2 = 0.63$) and *SSF* ($R^2 = 0.52$) have significant positive correlations under the MDS and the TDS, and there is no significant difference between $S_L$ and *SSF*. Therefore, we believe that using $S_{NL}$ under the MDS can evaluate the soil quality of rocky desertification areas more directly. However, soil quality in the karst ecosystem is affected by many factors, including natural factors (climate, rainfall, temperature), regional factors (topography, rock characteristics) and human activities [63,64]. Furthermore, plant community compositions and bedrock exposed rate are two dominating factors effects on soil quality in rocky desertification areas [34,38]; they indirectly affect SOM and enzyme activity to improve and affect soil quality during rocky desertification processes.

### 4.3. Relationship between Intense Rocky Desertification and Karst Shallow Fissures

There are two peculiar phenomena in our study area of rocky desertification. One phenomenon is that a large amount of bedrock is exposed, the surface soil layer is very thin and most of the soil is found in the karst shallow fissure; the other phenomenon is that arbores are mainly found above karst shallow fissure areas filled with soil, while only some low-biomass or short shrub vegetation can be grown around fissures [38,65] (Figure 8). According to the industry standard of the State Forestry Administration (LY/T 1840-2009) and the classification standard of rocky desertification [26,27], the two peculiar phenomena belong to the IRD area. However, TN, TP and SMC show a U-shaped variety in our IRD area and are consistent with the research of Chen et al. (2018) [66], which further shows that serious rocky desertification is not directly limiting soil quality [66,67]. These studies and our research show that the IRD area is not the area with the worst soil quality, and the characteristics of a high bedrock exposure rate, low vegetation coverage and narrow soil area may be the main influencing factors for the SQI. With the intensification of rocky desertification, the bedrock exposed rate increases, and the soil available area decreases. It is this high bedrock exposed rate that makes the soil exist in the low-lying groove, and the soil nutrients on the high terrain and bedrock were converged into the groove soil by rainwater scouring [2,68]. In addition, the nutrients brought by atmospheric nitrogen deposition were also collected in the soil. On the contrary, in areas with light rocky desertification grades, although the bedrock exposure rate is not obvious, there are more small-sized rocks in a large area of soil, which will be washed away by rain [68]. Therefore, the soil quality in areas with serious rocky desertification is better.

In rocky desertification areas, the soil quality and nutrients of karst shallow fissures (soil layer > 30 cm) are lower than those of surface soil (soil layer ≤ 30 cm) (Figure 8b), but they are more suitable for the growth of deep-rooted plants and can provide favorable habitats for these plants [2,38,69,70], which has great significance for karst ecological restoration. During the sample plot investigation, we found that the plant diversity is

not rich in the IRD area, and deep-rooted plants can be seen in the stone fissure soil. Our physicochemical properties were the highest in IRD, which is consistent with Lu et al. (2014) and Zheng et al., (2022) [68,71]. Therefore, we propose the question of whether the IRD area will develop into a large area of karst shallow fissure soil in the future (Figure 8).

## 5. Conclusions

We evaluated soil quality in different rocky desertification grades of southwest China using one model (SQI), two selection methods (MDS and TDS) and three scoring functions ($SSF$, $S_L$ and $S_{NL}$). Significantly different ($p < 0.05$) eco-environmental factors were found among the NRD, LRD, MRD and IRD. Except for TK, Mn and amylase, they all showed a U-shaped variety. The SQIs based on MDS showed IRD > NRD > LRD > MRD ($SSF$ and $S_L$) and IRD > NRD> MRD > LRD ($S_{NL}$). However, based on the TDS, the trend of soil quality is always IRD > NRD > MRD > LRD. Furthermore, we found that $S_{NL}$ ($R^2 = 0.63$, $p < 0.001$ and CV = 30.69%) was more suitable for the evaluation of soil quality in the KRD area. Overall, the IRD soil area is narrow, and the soil nutrients, enzyme activities and soil quality are not poor. We concluded that the limited soil area can provide an appropriate habitat for deep plant growth and that there is a tendency to develop into large-area karst shallow fissures in the IRD area of southwest China. In conclusion, the deepening of rocky desertification does not directly affect the soil quality in karst ecosystems, human disturbance, soil texture and topographic features may be the main factors. In particular, the increase in bedrock exposure rate is also a key factor affecting soil quality in rocky desertification areas. Our research is helpful for adopting a more appropriate strategy and practices for the restoration of different rocky desertification grade in the fragile karst areas of southwest China and provide scientific basis for soil quality management of degraded ecosystems.

**Author Contributions:** Q.W. and W.Z.: Conceptualization, Methodology, Software, Formal analysis, Data curation, Writing—original draft, Funding acquisition, Writing—review & editing, Visualization. W.Y. and E.W.: Methodology, Resources, Writing—review & editing, Supervision, Funding acquisition. C.R.: Software, Formal analysis. All authors have read and agreed to the published version of the manuscript.

**Funding:** This research was funded by "the young scientific and technological talents growth project of Guizhou Provincial Department of Education, grant number 2018KY320" and "the National Key R & D Program of China, grant number 2020YFA0608100".

**Acknowledgments:** The National Key R & D Program of China (Grant number 2020YFA0608100) and the young scientific and technological talents growth project of the Guizhou Provincial Department of Education (Grant number 2018KY320) provided financial support for this study. We gratefully acknowledge the in-kind support of National Engineering Laboratory for Applied Technology of Forestry and Ecology in South China, Central South University of Forestry and Technology, Changsha.

**Conflicts of Interest:** The authors declare no conflict of interest.

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
