# Peer review of "Soil Quality Assessment and Management in Karst Rocky Desertification Ecosystem of Southwest China"

_forests, doi:10.3390/f13091513_

Round 1
Reviewer 1 Report
Soil quality is a matter of high interest, as the production of goods and services depends on it. Human activities often contribute to soil degradation and the increase in quality indices in these areas is not of great interest from the point of view of crop production. For example, in the IRD area, the percentage of rocky outcrops on the surface is so high that we are discussing improving the quality of the soil which is practically non-existent!!!
Introduction
Good
Line 81 - delete "in conclusion"
Materials and Methods
If possible, place a map of the geographic location of the study área
Line 118 - How the points in each study plot were selected. How many points were sampled?
Line 128 - 129 - The Table where you indicate the methods is Table 3, which is in the Results section. I suggest that Table 3 move to the Material and Methods section. There is some mixing between Methods and Results, please do the separation.
I suggest you replace in Table 1 "thin" with "Shallow".
There are acronyms throughout the text that the meaning has not been indicated. Please verify and correct.
Results
In general, it is necessary to improve the quality of the Figures!
In my opinion, the use of acronyms in titles should be avoided
Figures and Tables should allow an autonomous reading, independent of the text, so all acronyms used should have the meaning in the captions
Linha 221 - sugiro substituir "in conclusion" por exemplo por "overall"
Figure 3 - What is the meaning of the capital and small letters in the graphic?
Figure 5 - Reading is a little confusing! Is it possible to improve? Add columns' color legend.
Lines 237-238 - is necessary to make reading numbers clearer!
Line 247 - it is not Table 3 that has this information!!
Lines 246-248 - check the sentence and correct!
Discussion
Good
Conclusion
Good
Author Response
Dear Editors and Reviewers:
Thank you for your letter and for the reviewers’ comments concerning our manuscript entitled “Soil quality assessment and management in karst rocky desertification ecosystem of Southwest China”(ID: forests-1916592). Those comments are all valuable and very helpful for revising and improving our paper, as well as the important guiding significance to our researches. We have studied comments carefully and have made correction which we hope meet with approval. Revised portion are marked in red in the paper. The main corrections in the paper and the responds to the reviewer’s comments are as follows:
Responds to the reviewer’s comments:
Reviewer #1:
- Response to comment:Line 81 - delete "in conclusion".
Response: We have deleted "in conclusion" in the manuscript.
- Response to comment: If possible, place a map of the geographic location of the study área.
Response: Thank you very much for this suggestion of the reviewer, which is very helpful to improve the quality of our manuscript. We have added the map of study area in the manuscript.
- Response to comment:How the points in each study plot were selected. How many points were sampled?
Response: In March 2021, soil sample plots of four different rocky desertification grades were selected and divided into three large 50 × 50 m plots. The distance between each quadrat was not less than 200 m. According to the characteristics of thin soil depths and high gravel content in the KRD areas, the sampled soil depths were 0-10 cm, 10-20 cm and 20-30 cm. Finally, soil samples were collected from depth of 0–10 cm, 10–20 cm, and 20–30 cm depths at five points (one point at the center and four points equidistant from the center toward the corner of the subplots) of each subplot. The five samples from each subplot were pooled to form a composite sample and the total number of soil samples collected was 4 treatments × 3 soil depths × 3 replicate sites × 5 subsamples = 180. Each sample contains three parts. One part was ring knife soil, and the fresh weight was weighed and brought back to the laboratory. One part was the soil enzyme activity sample. After sampling, put it into the liquid nitrogen box at - 20 ℃ and took back to the laboratory. In the laboratory, the soil enzyme activity sample was sieved to 2 mm and placed in a 4℃ refrigerator for further analysis. One part was the soil physical and chemical properties sample, taken back to the laboratory and air-dried and sieved to 2 mm and 0.149 mm for further analysis, respectively.
- Response to comment:The Table where you indicate the methods is Table 3, which is in the Results section. I suggest that Table 3 move to the Material and Methods section. There is some mixing between Methods and Results, please do the separation.
Response: We apologize for our negligence in not put the Table in the right place in the manuscript. We have adjusted in the manuscript according to the opinions of the reviewers.
- Response to comment:I suggest you replace in Table 1 "thin" with "Shallow".
Response: We have replace in Table 1 "thin" with "Shallow".
- Response to comment:There are acronyms throughout the text that the meaning has not been indicated. Please verify and correct.
Response: We apologize for our negligence in not complete the full name in the manuscript, which has been carefully checked and completed.
- Response to comment:In general, it is necessary to improve the quality of the Figures!
Response: Thank you very much for this suggestion of the reviewer, which is very helpful to improve the quality of our manuscript. We have improved the quality of the Figures in the manuscript.
- Response to comment:Result: In my opinion, the use of acronyms in titles should be avoided?
Response: Thanks again, we have deleted the use of acronyms in titles in the manuscript according your opinion.
- Response to comment:Figures and Tables should allow an autonomous reading, independent of the text, so all acronyms used should have the meaning in the captions.
Response: We are very sorry for our incorrect writing, we have checked and corrected it correctly.
- Response to comment:Lin 221 - sugiro substituir "in conclusion" por exemplo por "overall".
Response: We have replaced"in conclusion" to "overall".
- Response to comment:Figure 3 - What is the meaning of the capital and small letters in the graphic?
Response: We are very sorry for our incorrect writing. And we have added “Different capital letters represent different rocky desertification grades of the same soil layer have significant differences, and different small letters represent have significant differences between different soil layers of the same rocky desertification grade (P < 0.05)” in Figure 3.
- Response to comment:Figure 5 - Reading is a little confusing! Is it possible to improve? Add columns' color legend.
Response: Thank you very much for this suggestion of the reviewer, which is very helpful to improve the quality of our manuscript. We have try our best to improve Figure 5 in the revised manuscript.
- Response to comment:Lines 237-238 - is necessary to make reading numbers clearer!
Response: This is a very good suggestion, and we have worked hard to improve the clarity of the reading numbers.
- Response to comment:Line 247 - it is not Table 3 that has this information!!
Response: We are very sorry for our incorrect writing and have rewrite Table 5 to replace Table 3.
- Response to comment:Lines 246-248 - check the sentence and correct!
Response: We apologize for our negligence in the incorrect writing of our sentence. We have been revised to “Soil enzyme activity is greatly affected by pH, SMC, Mn and TK (Fig. 4). Sucrase was negatively correlated with pH and TK, and the correlation coefficients were -0.24 and -0.043, respectively. β-glucosidase was greatly affected by pH and the correlation coefficient was -0.025.” in manuscript.
Special thanks to you for your good comments: There are acronyms throughout the text that the meaning has not been indicated. Please verify and correct.
We tried our best to improve the manuscript and made some changes in the manuscript. These changes will not influence the content and framework of the paper. And here we did not list the changes but marked in red in revised paper.
We appreciate for Editors/Reviewers’ warm work earnestly, and hope that the correction will meet with approval.
Once again, thank you very much for your comments and suggestions.

Reviewer 2 Report
This study assessed the soil quality of four rocky desertification grades, namely, non rocky desertification, light rocky desertification, moderate rocky desertification, and intense rocky desertification. The soil quality index method was used to assess the soil quality based on 2 screening processes and by 3 weighting ways. I think estimating the 2 screening processes and 3 ways is an interesting topic and deserves to be explored. The study is quite simple and easy to follow. My major concerns including: 1) more information regarding material and method should be provided, 2) too many tables and figures which needed to be combined or deleted, 3) conclusions should consider bedrock exposure rate as the most important factor when evaluating soil quality of karst rocky desertification. My suggestion is minor revision.
Detailed comments:
L21: the full names of TK and Mn should be provided.
L26-27: not clear?
L27-31: Karst ecosystem is a special ecosystem, where rock desertification is common. Under the similar soil condition, SQI can be used to evaluate the soil quality. However, using SQI in evaluating soil quality of different degrees of rocky desertification may lead to wrong conclusions. For example, the SQI was the highest in IRD indicating intense rocky desertification is better than un-degraded karst soil. This is wrong. For this issue, the total soil mass and/or degree of rocky desertification (or bare rock rate) should be included in the calculation of SQI.
L109-113: more information should be provided. For example, the land use history, vegetation composition and diversity of different grades of rocky desertification.
L116-117: From the pictures, I think it was difficult to collect soil at a depth of 20-30cm. Because the soil appears to be very shallow.
L128: why Mn? Did the Mn have important environmental effects in his area?
L126-129: The analysis methods of the variables should be provided.
Fig2: S1, S2, S3 should be explained in the first sentence of the figure legend.
There were symbols by the fig 2g. delete?
The letters in the figure that indicating the multi-comparison results were not easy to follow.
L240: “Concentrations” is awkward
Fig3: all the abbrevations should be explained or replaced with their full names.
It was awkward that high enzyme activities found in the lowest soil depth, as well as the SOM and other soil physico-chemical properties. The S1S2S3 may be wrong denoted. Please check your results.
Fig4: why different colors and shapes? They needs to be explained in the figure legend.
What was the bar in the right side of the figure? From -1 to 1, there were same colors.
There was a symbol in the end of the figure.
Table1 and 2: combine them.
Table 3: Delete. It only needs to simply describe the soil analysis method in the method section.
L294: soil layers? I did not find any soil layer information in this figure.
What were the different colored symbols along with each bar in the fig5?
For S-NL in the Fig5a, please check the letters for the four bars.
L315-317: How was the fig 7 constructed, by what results and what method?
Table 4-6: were they important? Delete them?
L382-384: what do you mean?
L385-387: ???? I don’t get it.
Fig8: awkward figure and hypothesis. Please focus on your results.
L378-392: the IRD had better soil quality may mainly due to their limited soil mass that received high amount of N deposition and litter fall. In short, very little soil received large amount of resources, resulting in high nutrient concentrations and enzyme activity under the IRD condition. Therefore, the authors need to carefully generate their conclusion. The bedrock exposure rate need to be considered when using the SQI as indicators in this study.
L455: no results to support this part. Delete it.
L488-490: no evidence.
L490-491: total wrong. Karst rocky desertification has resulted serve soil degradation in southwest China. During past decades, ecological restoration improves the soil quality.
Any suggestions for policy makers should combine the SQI and bedrock exposure rate. More crucially, I think that the rate of bedrock exposure should be given more weight.
Author Response
Dear Editors and Reviewers:
Thank you for your letter and for the reviewers’ comments concerning our manuscript entitled “Soil quality assessment and management in karst rocky desertification ecosystem of Southwest China”(ID: forests-1916592). Those comments are all valuable and very helpful for revising and improving our paper, as well as the important guiding significance to our researches. We have studied comments carefully and have made correction which we hope meet with approval. Revised portion are marked in red in the paper. The main corrections in the paper and the responds to the reviewer’s comments are as follows:
Responds to the reviewer’s comments:
Reviewer #2:
- Response to comment:L21: the full names of TK and Mn should be provided..
Response: Thank you very much for your advice. We have provided the full name of TK(total potassium) and Mn(manganese) in manuscript.
- Response to comment:L26-27: not clear?
Response: We apologize for our negligence in not clear in the sentence, which has been carefully checked and revised to “However, the SQI was always in the trend of IRD > NRD > MRD > LRD based on the TDS” .
- L27-31: Karst ecosystem is a special ecosystem, where rock desertification is common. Under the similar soil condition, SQI can be used to evaluate the soil quality. However, using SQI in evaluating soil quality of different degrees of rocky desertification may lead to wrong conclusions. For example, the SQI was the highest in IRD indicating intense rocky desertification is better than un-degraded karst soil. This is wrong. For this issue, the total soil mass and/or degree of rocky desertification (or bare rock rate) should be included in the calculation of SQI.
Response: Thank you very much for this suggestion from the reviewer, the comment will greatly improve the quality of our articles and optimize the results. As you said, more factors should be considered in the calculation of soil quality in rocky desertification areas, such as bedrock exposure rate and vegetation coverage rate. However, when we used SQI model to calculate these indicators, they were all excluded from the dimensionality reduction analysis of principal component analysis, so they did not appear in the soil quality evaluation indicators of our manuscript.
- L109-113: more information should be provided. For example, the land use history, vegetation composition and diversity of different grades of rocky desertification.
Response: Thank you very much for this suggestion from the reviewer, the comment will greatly improve the quality of our articles and optimize the results. The land use history of the study area is relatively single, all of which are agricultural land and returning farmland to forests. We have also conducted a large number of surveys and statistics on plant diversity and community composition, and we will add and supplement according to your opinions. I hope I can get your approval.
- L116-117: From the pictures, I think it was difficult to collect soil at a depth of 20-30cm. Because the soil appears to be very shallow.
Response: As you can see from the picture, the soil layer in the rocky desertification area is very shallow, but it is OK to take 20-30cm through our efforts.
- L128: why Mn? Did the Mn have important environmental effects in his area?
Response: As you know, the calculation of soil quality is based on the dimensionality reduction analysis of a large number of indicators, and finally selects the appropriate full data set and the minimum data set for the calculation of soil quality. In our study, we measured soil biological, chemical and physical indicators under different levels of rocky desertification, including manganese, iron and a variety of elements, but finally manganese entered the full data set. Therefore, we chose manganese.
- L126-129: The analysis methods of the variables should be provided.
Response: We apologize for not providing a clear variable analysis method. According to your opinion, we have supplemented the variable analysis method. We hope to get your approval.
- Fig2: S1, S2, S3 should be explained in the first sentence of the figure legend.There were symbols by the fig 2g. delete? The letters in the figure that indicating the multi-comparison results were not easy to follow.
Response: We apologize for our negligence of not giving a detailed explanation. Thank you for your comments, which will greatly improve the quality of our manuscript. In Fig 2, there were no redundant symbols when we submitted it. It may be an error brought by the system. We have modified it according to your comment. We hope to get your approval.
- Fig3: all the abbrevations should be explained or replaced with their full names.It was awkward that high enzyme activities found in the lowest soil depth, as well as the SOM and other soil physico-chemical properties. The S1S2S3 may be wrong denoted. Please check your results.
Response: Thank you very much for having such an excellent reviewer as you. It is your serious and perfect opinions that will improve the quality of the manuscript. We have revised and supplemented the abbreviations according to your opinions, and also checked all the chart representations.
- Fig4: why different colors and shapes? They needs to be explained in the figure legend.What was the bar in the right side of the figure? From -1 to 1, there were same colors. There was a symbol in the end of the figure.
Response: We apologize for our negligence of not giving a detailed explanation. Thank you for your comments, which will greatly improve the quality of our manuscript. The different colors and shapes in Figure 4 mainly mean that we want to distinguish different levels of rocky desertification, so that readers can more intuitively see the differences between different levels of rocky desertification. The legend from - 1 to 1 indicates a difference between different indicators. There were no redundant symbols when we submitted it. It may be an error brought by the system. We have modified it according to your comment. We hope to get your approval.
- There was a symbol in the end of the figure.Table1 and 2: combine them. Table 3: Delete. It only needs to simply describe the soil analysis method in the method section.
Response: There were no redundant symbols when we submitted it. It may be an error brought by the system. According to your opinion, table 1 and table 2 have been merged and table 3 has been deleted.
- L294: soil layers? I did not find any soil layer information in this figure.What were the different colored symbols along with each bar in the fig5? For S-NL in the Fig5a, please check the letters for the four bars.
Response: We apologize for our negligence here. The different colors in Figure 5 are the soil quality scores of different rocky desertification grades. We have carefully checked the letters in Figure 5 according to your opinions. Thank you again for your opinions.
- L315-317: How was the fig 7 constructed, by what results and what method?Table 4-6: were they important? Delete them?
Response: Figure 7 is mainly to more intuitively reflect the sudden quality difference between the two methods. After calculating the soil quality through SQI model, a simple calculation of the soil quality percentage of different rocky desertification grades is carried out. Table 4 to table 6 are some specific details of calculation and evaluation of soil quality at different rocky desertification grade, which we think is necessary to retain. Thank you very much for your comments.
- L382-384: what do you mean?
Response: “Many studies have measured the concentration and distribution characteristics of the soil pH value, TN, TP, TK, SOM, some heavy metals and many soil enzymes in karst areas [34,42]. Some reported showed that except for soil pH values and Ca, which increased with rocky desertification grade, the soil component contents were MRD > LRD > IRD”, Through the citation of these references, we want to express that the soil properties in the rocky desertification area do not change regularly with the change of the rocky desertification grade, but have great differences with the different study plots. Thank you very much for your suggestion. I hope our explanation can be recognized by you.
- L385-387: ???? I don’t get it.
Response: Just like the previous proposal, what we want to express is that there are great differences due to climate factors, land use types, rock types and human interference.
- Fig8: awkward figure and hypothesis. Please focus on your results.
Response: Thank you very much for your suggestion. We have made appropriate modifications to Fig . 8 and carefully checked the results. We put forward this assumption because our results are very similar to the research results of karst shallow soil. Therefore, we put forward this assumption in the hope of your approval.
- L378-392: the IRD had better soil quality may mainly due to their limited soil mass that received high amount of N deposition and litter fall. In short, very little soil received large amount of resources, resulting in high nutrient concentrations and enzyme activity under the IRD condition. Therefore, the authors need to carefully generate their conclusion. The bedrock exposure rate need to be considered when using the SQI as indicators in this study.
Response: Thank you very much for this suggestion from the reviewer, the comment will greatly improve the quality of our articles and optimize the results. As you said, more factors should be considered in the calculation of soil quality in rocky desertification areas, such as bedrock exposure rate and vegetation coverage rate. However, when we used SQI model to calculate these indicators, they were all excluded from the dimensionality reduction analysis of principal component analysis, so they did not appear in the soil quality evaluation indicators of our manuscript.
- L455: no results to support this part. Delete it.
Response: Thank you very much for this suggestion from the reviewer. We have been deleted “Therefore, the soil indicators selected in this study are scientific and reasonable for the evaluation of soil quality in rocky desertification areas.”
- L488-490: no evidence.
Response: We agree with you very much. Indeed, there is no evidence to support our hypothesis. According to your opinion, we have deleted “Therefore, we propose the question of whether the IRD area will develop to a large area of karst shallow fissure soil in the future (Fig. 8)”.
- L490-491: total wrong. Karst rocky desertification has resulted serve soil degradation in southwest China. During past decades, ecological restoration improves the soil quality.
Response: We apologize for the mistake caused by our negligence. According to your opinion, we have deleted “Karst rocky desertification has resulted serve soil degradation in southwest China. During past decades, ecological restoration improves the soil quality” and hope this change to get your approval.
- Any suggestions for policy makers should combine the SQI and bedrock exposure rate. More crucially, I think that the rate of bedrock exposure should be given more weight.
Response: This is an extremely important suggestion, because it is particularly important for the quality of our manuscript. As you said, more factors should be considered in the calculation of soil quality in rocky desertification areas, such as bedrock exposure rate and vegetation coverage rate. However, when we used SQI model to calculate these indicators, they were all excluded from the dimensionality reduction analysis of principal component analysis, so they did not appear in the soil quality evaluation indicators of our manuscript.
Special thanks to you for your good comments: Any suggestions for policy makers should combine the SQI and bedrock exposure rate. More crucially, I think that the rate of bedrock exposure should be given more weight.
We tried our best to improve the manuscript and made some changes in the manuscript. These changes will not influence the content and framework of the paper. And here we did not list the changes but marked in red in revised paper.
We appreciate for Editors/Reviewers’ warm work earnestly, and hope that the correction will meet with approval.
Once again, thank you very much for your comments and suggestions.
Sincerely yours,
Wei Zheng
On behalf of all authors.
